# Running around the Country: An Analysis of the Running Phenomenon among Brazilian Runners

**DOI:** 10.3390/ijerph18126610

**Published:** 2021-06-19

**Authors:** Mabliny Thuany, Beat Knechtle, Thomas Rosemann, Marcos B. Almeida, Thayse Natacha Gomes

**Affiliations:** 1Centre of Research, Education, Innovation and Intervention in Sport (CIFI2D), Faculty of Sports, University of Porto, 4200-450 Porto, Portugal; mablinysantos@gmail.com; 2Medbase St. Gallen Am Vadianplatz, Vadianstrasse 26, 9001 St. Gallen, Switzerland; 3Institute of Primary Care, University of Zurich, 8091 Zurich, Switzerland; thomas.rosemann@usz.ch; 4Department of Physical Education, Federal University of Sergipe, São Cristóvão 49100-000, SE, Brazil; mb.almeida@gmail.com (M.B.A.); thayse_natacha@hotmail.com (T.N.G.)

**Keywords:** running, economic factors, sport events

## Abstract

Differences in economic and social aspects between Brazilian states/regions can determine participation in running. The purpose of this study was to identify the occurrence of the OUTrun (i.e., Out Running), mapping the main routes carried out by runners, as well as the factors associated with this behavior between different Brazilian regions. The sample comprised 1053 runners of both sexes (women: 426; men: 627) who answered an online questionnaire, providing information related to individual, socioeconomic (SES), and training characteristics. A logistic regression analysis was computed, considering the regions. South and Southeast regions received the largest number of runners; runners from the North and Northeast regions were those who left their states the most to compete. Factors related to the OUTrun were the preferred distance, SES, and age. The results provide information to facilitate access to running events and can provide benefits related to making the practice accessible to a larger number of people.

## 1. Introduction

Considered the main physical activity trend for the year 2020 [1], road running stands out among the most practiced sports in the world [2]. Data covering 96% of the United States’s race events, and 91% of the race results from Canada, Australia, and a portion of Africa, Asia, and South America between the years 1986 and 2018, showed an increase of 57.8% in the number of participants in these events [3]. In the Brazilian context, data covering races hosted in São Paulo and Rio de Janeiro showed an increase in participation and number of events during the last years [4]. Moreover, regarding economic factors, Brazilian data from 2019 showed that there were about 4 million runners who comprised a market of about BRL 3.1 billion/year in 2019 (about USD 12.4 billion/year) [5]. Individual costs are related to the purchase of sports equipment (e.g., shoes, Global Positioning System watches, clothing, etc.), knowledge about running (e.g., subscriptions to websites and specialized magazines), involvement in running groups/clubs, and especially participation in running events [5,6].

Despite the growth in the number of running events at a national level [7], the distribution of these events between the Brazilian states is not similar, with a huge discrepancy observed. The Southeast region concentrates the highest number of these events, as well as the most important and traditional events (e.g., São Silvestre, São Paulo Marathon, Rio de Janeiro Marathon, Pampulha International Tour, ten miles Garoto race) [8,9]. Therefore, there is a tendency for runners from the different Brazilian states going to this region to take part in these races, in a phenomenon that can be called “Out Running” (OUTrun), in which runners “exit” from their states of residence and “enter into” other places (states and/or countries) to participate in race events.

Although usually considered as a cheap and affordable practice, a deep look at the context of running reveals changes in the profile of its practitioners, and its “accessibility” can be questioned [10,11]. In general, studies showed that economic factors may be related to the involvement/practice in road running [12]. For example, in the Brazilian scenario, more than 600 races are held annually [13], whose registration prices can exceed BRL 200.00/person (about USD 35.00/person). Notwithstanding that this is not cheap or accessible for a large portion of the Brazilian population (and even for most runners), these events attract runners and usually reach their registration goals [14].

Economic and sociodemographic factors are also associated with running practice [12,15]. Moreover, individuals with higher socioeconomic status and education level and aged between 30 and 40 years are more likely to engage in the practice of the modality [2,12]. Thus, given the geographic extension of the country, which leads to differences in economic and social aspects between states/regions [16,17], the purpose of this study was to identify, between Brazilian regions, the occurrence of the OUTrun, mapping the main routes carried out by runners, as well as the factors associated with this behavior. Based on previous studies, we hypothesized that economic level must be associated with participation in running events.

## 2. Materials and Methods

### 2.1. Design and Sample

The present study is part of the InTrack Project [18], a cross-sectional research project developed with the aim to identify the predictors of runner’s performance in non-professional runners. To take part in the study, runners must have answered the online questionnaire, sent via social media, and available from September 2019 to March 2020, and be aged ≥18 years. From the 1173 answered questionnaires, 104 runners were excluded from the analysis because they did not report having taken part in any official race event in the last 12 months, and 16 were excluded because they did not answer all the mandatory questions. Thus, the final sample, for the present study, comprised 1053 runners of both sexes (women: 426; men: 627), from all the five Brazilian regions (Southeast: 409; Northeast: 350; South: 130; Midwest: 90; North: 74). This study was conducted in accordance with the Declaration of Helsinki and was approved by the Ethics Committee of the Federal University of Sergipe, Brazil (protocol n° 3.558.630).

### 2.2. Procedures and Data Collection

The information was collected through the online questionnaire “Profile characterization and associated factors for runner’s performance” [19]. The questionnaire allowed obtaining information related to (1) runner identification (age and sex), (2) anthropometric variables (height and weight), (3) sociodemographic profile (neighborhood, income, educational level, and marital status), (4) perception about the environment (natural or built) influence on the practice, (5) training variables (volume and frequency/week, sessions/day, practice time, pace (min/km), involvement in official races in the last 12 months, involvement in a running club, the use of a personal coach to guide the practice, motivation for the practice, and preferred distance), and (6) the family environment (family composition, family members engaged in running practice, involvement in sports during childhood, and family support for sporting involvement during childhood). The Intraclass Correlation Coefficient demonstrated that items were classified as “excellent” (R ≥ 0.75), and the Kappa Coefficient presented concordance “substantial” to “almost perfect”.

#### 2.2.1. Individual Characteristics

Sex, age, and state of residence were self-reported.

Anthropometric characteristics—height (cm) and body mass (kg)—were self-reported, as performed in previous studies [20]. Body mass index (BMI) was computed through the standardized formula (body mass (kg)/height (m)^2^).

Socioeconomic status (SES): runners were asked to provide an estimate of their monthly income, on a Likert scale, based on Brazilian minimum wage in 2019 [21]. Furthermore, answers were organized into two categories: “≤5 minimum wages” (≤4990.00 BRL or about ≤1.205 USD) and “>5 minimum wages” (>4990.00 BRL or about >1.205 USD).

Educational level: runners provided information regarding the highest education level achieved, being classified as “<elementary school”, “elementary school”, “high school”, or “university degree”.

#### 2.2.2. Training Characteristics

Running pace: participants were asked to provide information regarding their running pace (min/km) in their preferred distance.

Volume/km: the mean weekly value was reported (in kilometers) by runners.

Practice time: runners were categorized into two groups, namely “≤1 year” of practice and “>1 year” of practice.

Preferred distance: runners answered their preferred running distance, and they were categorized as “short distance” runners (until 10 km) or “long-distance” runners (above 10 km).

OUTrun phenomenon: information about runners’ participation in any race event in the last 12 months was provided by participants. We considered OUTrun when this participation occurred in a different state (or country) from the one of their residence.

### 2.3. Statistical Analysis

The information collected was presented as mean (and standard deviation) or frequency (%). Logistic regression analysis was computed to identify the predictors associated with the OUTrun phenomenon (chance to exit from their state of residence to participate in race events in another state or country), and models were built for each region. The variables included as predictors were sex (female or male), age, SES (“≤5 minimum wages”, or “>5 minimum wages”), and preference distance (“short distance” or “long distance”). For categorical variables, the first category reported was the reference group during data analysis. All the analyses were performed in SPSS 24.0 software (IBM Corp., Armonk, NY, USA), with a significant level set at 95%. The Gephi software (version 0.9.2) (Gephi, WebAtlas, Paris, France) was used to build the networks established by runners related to their travels between states (or countries).

## 3. Results

Table 1 presents descriptive information. The sample presented a mean running pace of 5:30 min/km and a mean BMI of 24.4 kg/m^2^. The mean age of the sample was 38.6 ± 8.4 years for women and 37.6 ± 9.9 years for men. The runners were from all states of the country, except Rondônia. Most of the runners reported living in the Southeast (38.2%) and Northeast (33.8%) regions.

The North, Northeast, and Midwest regions showed the highest percentage of runners who leave their states of residence to take part in races hosted in other states, and, except for the Northeast region, the majority of runners who traveled from their states to compete prefer long distances instead of short distances (Table 2). Taking sex into account, 68.1% and 57.1% of the women and men, respectively, reported preferring short distances.

Figure 1 presents the networks derived from the origin-destination dyad related to runner’s participation in racing events. In general, there was a greater outflow of runners from the states of São Paulo, Rio de Janeiro, Sergipe, Minas Gerais, Paraná, and Espírito Santo (indicated by the grey circles), while Santa Catarina, São Paulo, Rio de Janeiro, Rio Grande do Sul, Distrito Federal, and Espírito Santo states were those where runners arrived the most (coming from other states) to take part in running events (represented by the biggest circles). Furthermore, it is also possible to observe the runners’ “way out” to get involved in races in other countries, namely Argentina, Chile, Uruguay, the United States, Germany, Spain, France, Greece, and Portugal, highlighting the international facet related to the practice.

Table 3 presents the results of logistic regression analysis. Except for the South region, the preferred distance was a significant predictor for participation in the racing events in other states/countries, indicating that runners who preferred long-distance running tend to travel abroad to compete more than those who indicated short running distances as their preferred ones (Midwest: OR = 6.28, *p* = 0.017; North: OR = 10.10, *p* = 0.002; Northeast: OR = 1.95, *p* = 0.017; Southeast: OR = 7.02, *p* < 0.001). In addition, in the Southeast and Northeast regions, the higher the SES, the higher the chances of runners traveling abroad to take part in running events (Northeast: OR = 2.06, *p* = 0.017; Southeast: OR = 3.09, *p* = 0.002). Moreover, it highlights the role played by sex in the South region (OR = 0.32, *p* = 0.030) in the studied phenomenon (Table 3).

## 4. Discussion

The purpose of this study was to identify the occurrence of OUTrun, mapping the main routes carried out by runners, as well as the factors associated with this phenomenon. The main finding of the study was that the states of São Paulo, Rio de Janeiro, Sergipe, Minas Gerais, Paraná, and Espírito Santo are those with the highest number of runners who reported traveling from their states of residence to compete, while the states of Santa Catarina, São Paulo, Rio de Janeiro, Rio Grande do Sul, Distrito Federal, and Espírito Santo were those where runners arrived the most to take part in running events.

Results are associated with a large number of running events and some of the most important Brazilian road running races, which are hosted in South and Southeast regions, especially in the states of Rio de Janeiro, São Paulo, Rio Grande Sul, and Minas Gerais [13]. In a multilevel approach previously performed, it was observed that runners from states with more athletics events were those with the best performance [15]. Although the present study did not investigate differences in runners’ performance, an association between sports events, sports participation, and sports performance seems to exist [12]. Furthermore, a previous study conducted in Brazil reported these regions as having the friendliest environment, which may be related to a higher opportunity for citizens to take part in outdoor physical activities and training practice, which can lead to increases in demand for race events. In addition, as an outdoor event, long-distance running (half-marathon and marathon) performance tend to be associated with a natural environment (e.g., climate, weather, wind, altitude), and the North and Northeast Brazilian regions are those with the highest annual average temperature, meaning that this could impact in the runners’ choice in the place to compete.

About a quarter of the runners from the North, Northeast, and Midwest regions leave their states to participate in events in states from other regions, especially in long-distance running events. One possible reason is the fact that long-distance running races, such as marathons, are less frequent in some states. This can be observed in the calendar of running events for the year 2020, which indicated that of the 17 official marathons planned to occur in Brazil, only one of them was supposed to be hosted in the North region (Amazonas) and five in the Northeast region, while the state of São Paulo was supposed to host at least two of them [22].

In this context, an increase in the number of runners participating in long-distance running, namely half-marathon running, has been observed worldwide [23]. Moreover, the profile of these runners has aroused the attention of the tourism market (“Maratourism”), with several companies providing logistic advisory services to runners, helping them in their movement to the place of the competition, as well as providing accommodation options and, eventually, leisure entertainment during the time the runner spends in the city of the event. This service moves the local economy in several segments: hotel chain, transportation, restaurants, and business [5].

As hypothesized, SES was associated with the OUTrun. However, this result was only observed in runners from the Southeast and Northeast regions, where individuals with higher monthly incomes were more likely to participate in events outside their state of residence. According to Berger et al. [24], adherence to a given behavior/practice is associated with individual economic issues, in association with the available time and the personal feeling about the relevance of the sports practice. This information confirms findings from previous studies. For example, a previous study, investigating the demographic and economic model in different practices (sports participation, cycling, swimming, running, fitness, gymnastics, walking, football, dancing, and tennis) showed that the model can be used to predict participation in several sports. Especially in running, higher economic level and working time were associated with involvement in the modality [12,15]. Furthermore, evidence suggests an inverse relationship between event registration fees (registration fees and other associated costs) and the number of participants in the events [11], in opposition to the idea that the modality is accessible to everyone, regardless of their economic conditions, for example [10].

The relevance of economic conditions in running practice is further reinforced when it is noted that the registration fee in a single event of the modality (with an average registration fee of BRL 101/person or USD 19.33) could represent about 10% of the Brazilian minimum wage in the year 2020 [21]. In addition, other costs related to involvement in the events cannot be disregarded, which may become a barrier to participation in running events. As mentioned above, most long-distance running events require runners to travel from their state of residence to take part in the events, which demands high costs, in association with the higher registration fees. In the present study, runners who indicated long-distance running races as their preference were more likely to leave their states to compete.

Despite the reduction in the ratio between sexes over the years, the male frequency is still higher in absolute terms than the female in competitions [7,25]. Although in the present study men reported a greater preference for long-distance running events than women, sex was found to be a significant predictor of the occurrence of OUTRun only in the South region, where men were less likely to compete in other states/countries than women. Since male runners seemed to prefer long-distance running events, the fact that the South region hosts some Brazilian half-marathon, marathon, and ultramarathon races, runners who deserve to take part in events in such distances do not need to travel abroad to perform.

Although the increase in the number of participants in the modality could be seen throughout the country [4], there were differences regarding the practice preference between the regions. For example, in the Midwest Region, about 8% of the adults indicated running as their preferred sports practice, while in the South Region, only 3.6% of the adults indicated such a preference [26]. Even so, the South region receives a large number of runners from other states, who usually travel to this region to take part in the marathons held in Porto Alegre and Florianopolis, which, due to the topography of the area where they occur, are considered some of the best races for professional runners obtain their best scores [27,28,29], which is highlighted by the high frequencies of the best times achievement, among Brazilian marathoners, observed in these events (61% for men and 51% for women).

Regarding the limitations of the present study, it is worth mentioning differences in the sample size between the regions; however, the purpose of the study was not to obtain a representative sample in all states or regions. In addition, questions related to the individual investment in practice could help to elucidate issues related to individual and local economic aspects. Despite the previous validation of the instrument used, and the fact that the online survey has been used in previous studies, we are aware of the bias associated with sample selection, due to the strategy used to publicize the questionnaire. Given this limitation, we suggest caution in the generalization of the information. Finally, to our knowledge, this is the first study that highlights the main networks established between road running and the factors associated with OUTrun. Future studies should compare performance at races with different costs of participation (e.g., free and paid events) and should investigate states variables (e.g., economic, cultural, social, and tourist potential characteristics) that are possibly associated with race events distribution, as well as differences in performance between athletes in different races.

## 5. Conclusions

OUTrun was more frequent in runners from São Paulo (Southeast), Rio de Janeiro (Southeast), Sergipe (Northeast), Minas Gerais (Southeast), Paraná (South), Espírito Santo (Southeast), and Rio Grande do Sul (South), while Santa Catarina (South), São Paulo, Rio de Janeiro, Rio Grande do Sul, Distrito Federal (Midwest), and Espírito Santo were the states that received most of the runners. Regarding the factors associated with OUTrun, different variables were significantly associated with it for different regions. Except for the South region, the distance preference (long distance) was shown to be a significant predictor, increasing the chances of runners traveling to participate in these events in another city/country. Economic level was also associated with OUTrun in Northeast and Southeast regions, while sex was a significant predictor in the South region. The present study can be used by stakeholders and sports events organizers to develop strategies to make these events more accessible, and allowing the benefits associated with the sport practice to reach a larger number of people, regardless of their SES or place of residence.

## Figures and Tables

**Figure 1 ijerph-18-06610-f001:**
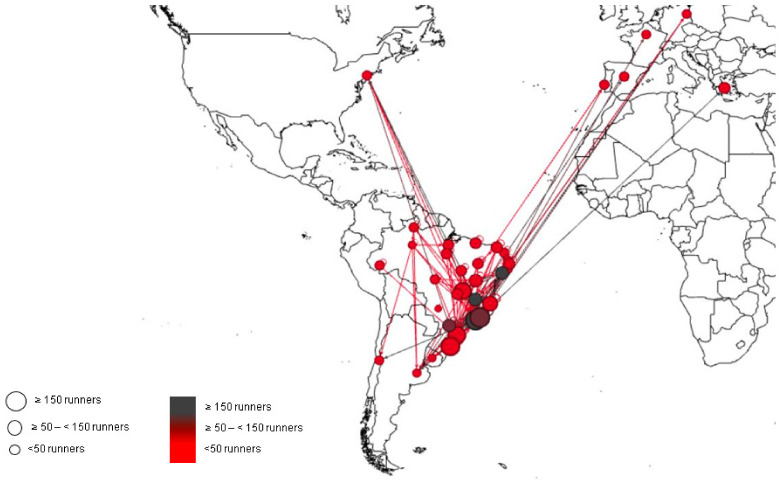
Network derived from runners exit and entrance in states (or countries) to take part in road running events.

**Table 1 ijerph-18-06610-t001:** Descriptive statistics of runners, by region.

Variables	Brazil	Midwest (*n* = 90)	Northeast (*n* = 350)	North (*n* = 74)	Southeast (*n* = 409)	South (*n* = 130)
Sex						
Woman	426 (40.5%)	41 (45.6%)	135 (38.6%)	32 (43.2%)	160 (39.1%)	58 (44.6%)
Man	626 (59.4%)	49 (54.4%)	214 (61.1%)	42 (56.8%)	249 (60.9%)	72 (55.4%)
Age (years)	38.0 (9.3)	39.2 (9.7)	37.8 (9.9)	35.6 (10.4)	38.6 (8.9)	37.6 (8.6)
BMI (kg.m^2^)	24.4 (3.3)	24.0 (3.3)	24.4 (3.2)	25.1 (3.8)	24.6 (3.3)	24.0 (3.1)
Running pace (min/km)	5:30 (1:09)	05:38 (1:19)	05:38 (1:19	5:38 (1:26)	05:28(1:06)	05:14 (55.7)
Volume/km	35.4 (29.5)	36.0 (27.4)	31.1 (26.1)	34.1 (21.6)	39.7 (35.1)	34.5 (22.5)
SES						
≤5 minimum wages	552 (52.4%)	32 (36%)	179 (51.1%)	45 (62.5%)	242 (59.4%)	55 (43.7%)
>5 minimum wages	485 (46.1%)	57 (64%)	166 (47.4%)	27 (37.5%)	164 (40.1%)	71 (56.3%)
Educational level						
<Elementary school	8 (0.8%)	1 (1.1%)	1 (0.3%)	0	6 (1.5%)	0
Elementary school	42 (4.0%)	4 (4.4%)	10 (2.9%)	3 (4.1%)	23 (5.6%)	2 (1.5%)
High school	239 (22.7%)	13 (14.4%)	78 (22.3%)	18 (24.3%)	107 (26.2%)	23 (17.7%)
University degree	758 (72.0%)	70 (77.8%)	261 (74.6%)	52 (70.3%))	271 (66.3%)	104 (80.0%)
Missing	6 (0.6%)	2 (2.2%)		1 (1.4%)	2 (0.5%)	1 (0.8%)
Practice time						
Until 1 year	167 (15.9%)	16 (17.8%)	65 (18.6%)	11 (14.9%)	70 (17.1%)	5 (3.8%)
>1 year	886 (84.1%)	74 (82.2%)	285 (81.4%)	63 (85.1%)	339 (82.9%)	125 (96.2%)

**Table 2 ijerph-18-06610-t002:** Percentage of runners who reported to travel abroad their states of residence to take part in running events, by region and distance of preference.

Region	Total *	Distance of Preference		
		Short Distances **	Long Distances **	Missing **
Southeast	54 (13.4%)	9 (17.0%)	44 (81.5%)	1 (1.9%)
Northeast	82 (24.5%)	41 (50.0%)	40 (48.8%)	1 (1.2%)
North	18 (25.5%)	6 (33.3%)	12 (66.7%)	
Midwest	21 (24.4%)	4 (19.0%)	17 (81.0%)	
South	23 (18.1%)	11 (47.8%)	12 (52.2%)	

* Percentage values based on the total runners in each region; ** percentage values based on the number of runners who left their states to compete.

**Table 3 ijerph-18-06610-t003:** Results from the logistic regression for the predictors associated with traveling abroad to take part in race events.

Variables	Midwest	North	Northeast	Southeast	South
	OR (95% CI)				
Sex (male)	0.70 (0.18–2.76)	1.72 (0.39–7.70)	1.49 (0.84–2.66)	0.64 (0.32–1.29)	0.32 (0.12–0.90) *
Age (years)	0.97 (0.90–1.05)	1.08 (0.99–1.17)	1.02 (0.99–1.05)	1.00 (0.96–1.04)	0.97 (0.91–1.04)
SES (>5 minimum wages)	5.18 (0.49–55.31)	1.57 (0.33–7.42)	2.06 (1.14–3.72) *	3.09 (1.50–6.39) *	1.56 (0.53–4.57)
Distance (long distance)	6.28 (1.38–28.53) *	10.10 (2.33–43.83) *	1.95 (1.13–3.38) *	7.02 (2.97–16.63) *	2.67 (0.98–7.33)

*p* < 0.05 *.

## Data Availability

Data are available from the authors upon request.

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
