# Peer review of "Running around the Country: An Analysis of the Running Phenomenon among Brazilian Runners"

_ijerph, 2021, doi:10.3390/ijerph18126610_

Round 1
Reviewer 1 Report
Dear Authors,
You present an interesting study. I really appreciated that you highlight the behavior of the runners. On the toher hand, I think it is necessary to broaden the introduction, including a complete theoretical framework in order to highlight the importance of the study. For example, to include general data on employment, income, etc. of the different areas (north, south…) or information about the number of athletic license of the country. Furthermore, I miss references to previous studies that have analyzed similar variables, although in other territories.
Regarding material and methods, more information about the scales and items used in the questionnaire is appreciated (point 2.2).
I would like to highlight the figure 1. It is very explanatory, so it is deserves special attention in the discussion and conclusion sections.
In the discussion section, it would be interesting to emphasize that the runners must travel to other territories to participate in the races. Why there are no events in these areas? Lack of political and social support? Lack of good transport links?
Finally, it would be positive to include future lines of research in the conclusions section. For example: The future intention of participating in a running event in an area where they are not currently taking place, could be studied.
It is necessary to review the reference list.
Author Response
Reviewer comment:
Dear Authors,
You present an interesting study. I really appreciated that you highlight the behavior of the runners. On the toher hand, I think it is necessary to broaden the introduction, including a complete theoretical framework in order to highlight the importance of the study. For example, to include general data on employment, income, etc. of the different areas (north, south…) or information about the number of athletic license of the country. Furthermore, I miss references to previous studies that have analyzed similar variables, although in other territories.
Authors’ answer: We the comments/suggestions, and we tried to answer them, improving the quality of the manuscript.
Reviewer comment: Regarding material and methods, more information about the scales and items used in the questionnaire is appreciated (point 2.2).
Authors’ answer: We appreciate this suggestion. Information can be checked in the published paper (regarding the validation of the instrument), but we included a summary in the methods section.
Reviewer comment: I would like to highlight the figure 1. It is very explanatory, so it is deserves special attention in the discussion and conclusion sections.
Authors’ answer: We appreciate this suggestion. More information was presented and discussed.
Reviewer comment: In the discussion section, it would be interesting to emphasize that the runners must travel to other territories to participate in the races. Why there are no events in these areas? Lack of political and social support? Lack of good transport links?
Authors’ answer: Thanks for your comment. We included more information in the discussion section.
Reviewer comment: Finally, it would be positive to include future lines of research in the conclusions section. For example: The future intention of participating in a running event in an area where they are not currently taking place, could be studied.
Authors’ answer: Thanks for your comment. We include other potential suggestions for future studies.
Reviewer comment: It is necessary to review the reference list.
Authors’ answer: Adjusted.
Reviewer 2 Report
The authors have presented a study / evaluation of a survey on the distribution of running events in Brazil.
The survey is methodically okey, even if only participants via the social networks were addressed. The number of participants seems to be representative, even if no sample size or strength of effects is calculated.
the presentation of the results and statistical considerations are comprehensible and coherent. so is the discussion of the results.
The derivation of the results is well founded, even if the results are not particularly surprising.
Author Response
Reviewer comment: The authors have presented a study / evaluation of a survey on the distribution of running events in Brazil.
The survey is methodically okey, even if only participants via the social networks were addressed. The number of participants seems to be representative, even if no sample size or strength of effects is calculated.
Authors’ answer: We appreciate the reviewer comments. Regarding the strategy used to collect data, this is an usual strategy, used in previous studies (Janssen et al., 2020; Nikolaidis & Knechtle, 2020), but we included a note about it in the limitation section.
Reviewer comment: The presentation of the results and statistical considerations are comprehensible and coherent. so is the discussion of the results.
The derivation of the results is well founded, even if the results are not particularly surprising.
Authors’ answer: We appreciate the reviewer comments, we tried to improve the quality of results presentation, presenting more details.
Reviewer 3 Report
Dear editor and authors, firstly, thanks for the opportunity to review this manuscript. I have some comments that need to be clarified and / changed in the text:
- I suggest standardizing in the text and tables the same measures used throughout the manuscript. The changes from min/km to s / km during the reading were not legal.
- In table 1, the "." of the Midwest's BMI is in the wrong place.
- I think it would be interesting to add the last column in table 1, with the data from all regions grouped, so that we could visually compare the regions with themselves, and with Brazil.
- Table 2: if only these corridors correspond to the objective of the study, why does table 1 have other corridors that are not part of the sample to be studied?
- About the regression, was it performed with the information in table 1 or 2?
- There are many data presented as collected in the methodology that is not presented in the sample characterization and can influence the analysis, such as the educational level of the population, which is even mentioned in the 4th paragraph of the discussion. I suggest presenting all the data.
- It's mentioned more than once that the educational level can influence results, but this variable was not added in the regression, and it isn't clear why this happened.
- In some points, a difference is mentioned between men and women, which weren't presented, either in a table, figure or text. For example, see the excerpt below (taken from the discussion):
"Although in the present study men reported a greater preference for long-distance running events than women, the sex was found to be a significant predictor to the occurrence of OUTRun only in the South region where men were more likely to compete in other states/countries then women."
Where was this preference of men for long-distance placed? I didn't find it in the text. Again, I suggest that all data should be presented.
- Only the central-west and south regions are in agreement with the hypothesis of the study, however, this was not presented and discussed as it should.
- The discussion must be improved.
- About the conclusion, since none of the variables were significant for all regions, instead of generalizing a result that cannot be applied, wouldn't it be better to assume/show the results for regression by region?
Author Response
Dear editor and authors, firstly, thanks for the opportunity to review this manuscript. I have some comments that need to be clarified and / changed in the text:
Reviewer comment: I suggest standardizing in the text and tables the same measures used throughout the manuscript. The changes from min/km to s / km during the reading were not legal.
Authors’ answer: Adjusted.
Reviewer comment: In table 1, the "." of the Midwest's BMI is in the wrong place.
Authors’ answer: Thanks for the comment. It was corrected.
Reviewer comment: I think it would be interesting to add the last column in table 1, with the data from all regions grouped, so that we could visually compare the regions with themselves, and with Brazil.
Authors’ answer: Information included.
Reviewer comment: Table 2: if only these corridors correspond to the objective of the study, why does table 1 have other corridors that are not part of the sample to be studied?
Authors’ answer: We thanks the reviewer for the question. Indeed, in table 1, we presented data from all the subjects who took part in the study, while in table 2 we described the frequency of runners who reported to travel abroad their state of residence to take part in race events, and that is the reason why the subjects numbers are different in both tables.
Reviewer comment: About the regression, was it performed with the information in table 1 or 2?
Authors’ answer: Based on our aim, we performed the regression analysis taking into account the whole sample. Using the logistic regression, our dependent variable was categorical – travel abroad to compete, or not travel abroad to compete. Taking this into account, would not make sense only to consider subjects described in table 2.
Reviewer comment: There are many data presented as collected in the methodology that is not presented in the sample characterization and can influence the analysis, such as the educational level of the population, which is even mentioned in the 4th paragraph of the discussion. I suggest presenting all the data.
Authors’ answer: We appreciate this comment. In the methods section we included in section 2.2 “Procedures and data collection” all information available through the use of the questionnaire (runner’s identification; anthropometric variables; sociodemographic profile; perception about the environment influence on the practice; training variables; and the family environment). However, we did not include educational level (and other information) in statistical analysis. In despite of the fact that education level can be associated with SES, we decided to use only the income in the analysis. In the discussion section, we included the information just to illustrate (and even trying to explain results fond), since it was associated with the results reported by Breuer et al., 2013.
Reviewer comment: It's mentioned more than once that the educational level can influence results, but this variable was not added in the regression, and it isn't clear why this happened.
Authors’ answer: Educational levels was not included in the analysis due to its association with income. Some sentences in the discussion section were adjusted.
Reviewer comment: In some points, a difference is mentioned between men and women, which weren't presented, either in a table, figure or text. For example, see the excerpt below (taken from the discussion):
"Although in the present study men reported a greater preference for long-distance running events than women, the sex was found to be a significant predictor to the occurrence of OUTRun only in the South region where men were more likely to compete in other states/countries then women."
Authors’ answer: Thanks for the comment. We included this information in results section.
Reviewer comment: Where was this preference of men for long-distance placed? I didn't find it in the text. Again, I suggest that all data should be presented.
Authors’ answer: Information was included in the results section.
Reviewer comment: Only the central-west and south regions are in agreement with the hypothesis of the study, however, this was not presented and discussed as it should.
Authors’ answer: Thanks for the comment. We tried to discuss about these results.
Reviewer comment: The discussion must be improved.
Authors’ answer: We appreciate this comment. We tried to improve the discussion, based on reviewer’s comments/suggestions.
Reviewer comment: About the conclusion, since none of the variables were significant for all regions, instead of generalizing a result that cannot be applied, wouldn't it be better to assume/show the results for regression by region?
Authors’ answer: We adjusted the conclusion, based on the reviewer comment.
Reviewer 4 Report
GENERAL COMMENTS
The topic of the paper is interesting and fits the scope of the journal. The text is relatively well written and composed. Please find below severe minor comments.
SPECIFIC COMMENTS
Line 67. Please refer the inclusion and exclusion criteria of this study.
Table 1. Please add the units of each parameter.
Table 1. BMI Midwest: Please replace "240." with "24.0"
Lines 195-197. Please refer more details about two studies and after compare these studies with the present study (10, 28).
Lines 214. Why do you believe that men had a greater preference for long distance running than women? In my opinion because women are less competitive, training less and wanting to train less.
Author Response
GENERAL COMMENTS
The topic of the paper is interesting and fits the scope of the journal. The text is relatively well written and composed. Please find below severe minor comments.
SPECIFIC COMMENTS
Reviewer comment: Line 67. Please refer the inclusion and exclusion criteria of this study.
Authors’ answer: We appreciate this suggestion. The information is presented in the methods section: “To take part in the study, runners must have answered the online questionnaire, sent via social media, and available from September 2019 to March 2020, and be aged ≥18 years. From the 1173 answered questionnaires, 104 runners were excluded from the analysis because they did not report having taken part in any official race event in the last 12 months, and 16 were excluded because they did not answer all the mandatory questions”.
Reviewer comment: Table 1. Please add the units of each parameter.
Authors’ answer: Information included.
Reviewer comment: Table 1. BMI Midwest: Please replace "240." with "24.0"
Authors’ answer: Adjusted.
Reviewer comment: Lines 195-197. Please refer more details about two studies and after compare these studies with the present study (10, 28).
Authors’ answer: Thanks for this comment. We included more information in this sentence.
Reviewer comment: Lines 214. Why do you believe that men had a greater preference for long distance running than women? In my opinion because women are less competitive, training less and wanting to train less.
Authors’ answer: It is an interesting question. Indeed, women trend to present a profile less competitive than men, but this does not mean they train less or want to train less –a huge number of sociocultural factors not provide to women the same opportunities to take part in sport as given to men. So:
- only in the 1984, women were allowed to compete in race events. It means that “be competitive” did not matter to woman compete. And these restrictions led to the adoption of some behaviours that may echo until nowadays;
- moreover, to take part in long-distance events, runners must to spend more time in training, and we also know that, in general, women have lower available time than men, given they have to work, and also spend time in domestic chores (and even take care of children) – and these activities are not always share with their male partners (in Brazilian context, this is also true);
- in addition, another point that must be considered, is related to the existence of a friendly environment to training. Security is not perceived in the same way for both sexes; and woman are more prone to be victim of violence than men. It means that to training for long distances, runners must have to cover, also, long distances and even to use routes not so “safe”, especially for woman. And sure, this can reduce female involvement in long-distance trainings.
- if you look some data regarding the female runners’ involvement in long-distance races, it is possible to observe that, proportionally, there was a higher increase in the number of women who take part in marathon events than men, between 2008-2014 (an increase of 21.61% among men; and an increase of 33.35% among women);
- finally, even the female performance is not still higher than male (and physiological aspects can explain this difference), when compared both sexes, regarding how good they are in keeping their pace in the marathon, male runners showed to be 18.61% less good at keeping an even pace – highlighting that female runner commitment with training is far to be lower than their male peer.
So, taking these factors into account, we believe that this difference regarding the preferred distances between sexes, is much more related to sociocultural aspects, than to a lower commitment, a less competitive profile, or a deserve to “training less” of women.
Reviewer 5 Report
The manuscript titled "Running Around the Country: An Analysis of the Running Phenomenon Among Brazilian Runners" is an interesting peace of research. The paper no requires to much modifications. Introduction, methodology, results and discussion are interesting and well elaborated. The sample size's is acceptable and the methodological approach to the investigation attend the expectations of the Journal. Regarding to the conclusion the manuscript present a reasonable conclusion. Nonetheless, despite of the result is expect a recommendation for future investigations. Please, consider as a suggestion to include a recommendation in the last part of the conclusion.
Author Response
Reviewer comment: The manuscript titled "Running Around the Country: An Analysis of the Running Phenomenon Among Brazilian Runners" is an interesting peace of research. The paper no requires to much modifications. Introduction, methodology, results and discussion are interesting and well elaborated. The sample size's is acceptable and the methodological approach to the investigation attend the expectations of the Journal. Regarding to the conclusion the manuscript present a reasonable conclusion. Nonetheless, despite of the result is expect a recommendation for future investigations. Please, consider as a suggestion to include a recommendation in the last part of the conclusion.
Authors’ answer: We appreciate the comments/suggestion, and we included recommendations at the end of the discussion.
Round 2
Reviewer 1 Report
Dear Authors,
Thank you for your response. I believe that the paper has really improved.
In order to broaden this line of research, I encourage you to continue with the study.
Author Response
Dear Authors,
Thank you for your response. I believe that the paper has really improved.
In order to broaden this line of research, I encourage you to continue with the study.
Answer: thank you for your positive comment
Reviewer 3 Report
Reviewer comment: Was it performed with the information in table 1 or 2?
Authors’ answer: Based on our aim, we performed the regression analysis taking into account the whole sample. Using the logistic regression, our dependent variable was categorical – travel abroad to compete, or not travel abroad to compete. Taking this into account, would not make sense only to consider subjects described in table 2.
New reviewer comment: I have to disagree with this statement. Since within the objective of the study it is described that they intend to "mapping the main routes carried out by runners, as well as the factors associated with this behavior", it makes complete sense to analyze only those who traveled to compete. Note that I'm not saying that everyone's analysis is wrong, however, it can complement the analysis of those who traveled, to verify whether or not there is a similar behavior between traveling or not traveling. Anyway, I suggest including this new analysis.
Reviewer comment: There are many data presented as collected in the methodology that is not presented in the sample characterization and can influence the analysis, such as the educational level of the population, which is even mentioned in the 4th paragraph of the discussion. I suggest presenting all the data.
Authors’ answer: We appreciate this comment. In the methods section we are included in section 2.2 "Procedures and Data Collection" all information available through the use of the questionnaire (runner's identification; anthropometric variables; sociodemographic profile; perception about the environment influence on the practice; training variables; and the family environment ). However, we did not include educational level (and other information) in statistical analysis. In spite of the fact that education level can be associated with SES, we decided to use only the income in the analysis. In the discussion section, we included the information just to illustrate, since it was associated with the results reported by Breuer et al., 2013.
New reviewer comment: I understand that you have not used this data in the statistical analysis. However, I emphasize that these data must be presented clearly to readers and evaluators, as from them we can make our inferences and/or comparisons with other studies. For example comparisons and/or inferences whether or not anthropometry is in agreement with what was found in other studies, or whether this sample is a special population (eg, overweight). As for the level of education, we can have inferences about the high or low level, according to regions and other studies. Anyway, in addition to simply saying that they were collected and classified as excellent, they must present the data. That's why I again suggest inserting them clearly in the manuscript.
Author Response
Reviewer comment: Was it performed with the information in table 1 or 2?
Authors’ answer: Based on our aim, we performed the regression analysis taking into account the whole sample. Using the logistic regression, our dependent variable was categorical – travel abroad to compete, or not travel abroad to compete. Taking this into account, would not make sense only to consider subjects described in table 2.
New reviewer comment: I have to disagree with this statement. Since within the objective of the study it is described that they intend to "mapping the main routes carried out by runners, as well as the factors associated with this behavior", it makes complete sense to analyze only those who traveled to compete. Note that I'm not saying that everyone's analysis is wrong, however, it can complement the analysis of those who traveled, to verify whether or not there is a similar behavior between traveling or not traveling. Anyway, I suggest including this new analysis.
Authors’ answer: Once more, we thank the reviewer for the suggestion. But when we performed the logistic regression analysis, we also identified the predictors related to runners traveling to compete. As our dependent variable was categorial (“travel” or “not travel”), our results highlight the increases in odds to “traveling to compete” based on variables predictors inserted in the model.
Moreover, if we only consider those runners who “travelled to compete”, we only would have one group, whose dependent variable would not be expressed as categorical. So, would not be possible to investigate the predictors to “traveling abroad”, since all of them reported this behaviour. And also, would not be possible to perform some group differences analysis, because all the subjects would belong to the same group.
We hope the reviewer understand our explanation and our point of view.
Reviewer comment: There are many data presented as collected in the methodology that is not presented in the sample characterization and can influence the analysis, such as the educational level of the population, which is even mentioned in the 4th paragraph of the discussion. I suggest presenting all the data.
Authors’ answer: We appreciate this comment. In the methods section we are included in section 2.2 "Procedures and Data Collection" all information available through the use of the questionnaire (runner's identification; anthropometric variables; sociodemographic profile; perception about the environment influence on the practice; training variables; and the family environment ). However, we did not include educational level (and other information) in statistical analysis. In spite of the fact that education level can be associated with SES, we decided to use only the income in the analysis. In the discussion section, we included the information just to illustrate, since it was associated with the results reported by Breuer et al., 2013.
New reviewer comment: I understand that you have not used this data in the statistical analysis. However, I emphasize that these data must be presented clearly to readers and evaluators, as from them we can make our inferences and/or comparisons with other studies. For example comparisons and/or inferences whether or not anthropometry is in agreement with what was found in other studies, or whether this sample is a special population (eg, overweight). As for the level of education, we can have inferences about the high or low level, according to regions and other studies. Anyway, in addition to simply saying that they were collected and classified as excellent, they must present the data. That's why I again suggest inserting them clearly in the manuscript.
New Authors’ answer: Included.